# Smoothing Slot Attention Iterations and Recurrences

**Rongzhen Zhao** [1]  **Wenyan Yang** [1]  **Juho Kannala** [2 3]  **Joni Pajarinen** [1]

## Abstract

Slot Attention (SA) lies at the heart of mainstream Object-Centric Learning (OCL). Image features can be aggregated into object-level representations by SA *iteratively* refining cold-start query slots. For video, such aggregation proceeds by SA *recurrently* shared across frames, with queries cold-started on the first frame while transitioned from the previous frame's slots thereafter. However, cold-start queries lack sample-specific cues thus hindering precise aggregation on image or video's first frame; Non-first frames' queries are already sample-specific thus requiring aggregation transforms different from the first frame. We address these issues with our *SmoothSA*: (1) To smooth SA iterations on image or video's first frame, we *preheat* cold-start queries with rich input-feature information, by a tiny module self-distilled inside OCL; (2) To smooth SA recurrences across video's first and non-first frames, we *differentiate* the homogeneous aggregation transforms by using full and single iterations respectively. Comprehensive experiments on object discovery, recognition and visual reasoning validate our method's effectiveness. Further visual analyses illuminate the underline mechanisms. Our *source code*, *model checkpoints* and *training logs* are provided on https://github.com/Genera1Z/SmoothSA.

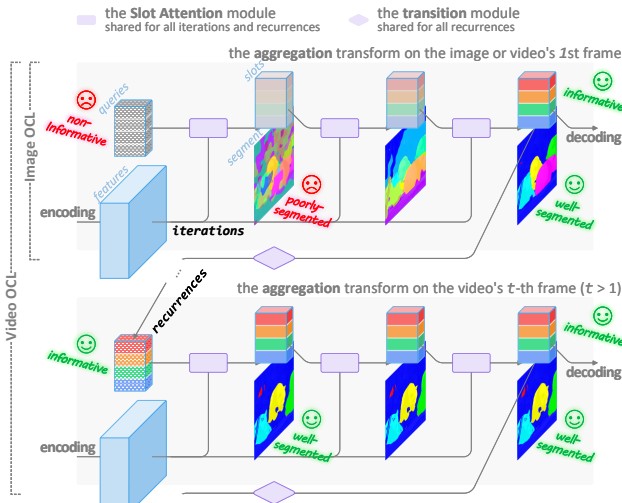

*Figure 1.* Image Object-Centric Learning (OCL) is realized via Slot Attention (SA) *iterations* on image, while video OCL is via SA *recurrences* across video frames. In SA iterations on image or video's first frame, the intrinsic **cold-start queries** lack information for accurate aggregation; In SA recurrences across video's first and non-first frames, the mainstream **homogeneous transforms**, i.e., the identical three SA iterations, cannot jointly adapt to the first and non-first queries, which have a significant information gap.

## 1. Introduction

Object-Centric Learning (OCL) (Locatello et al., 2020) aims to represent objects (and the background) in a visual scene as distinct vectors. Such structured and compact representation

can outperform dense feature maps in advanced vision tasks. Evolving object slots over time captures more accurate dynamics modeling (Villar-Corrales & Behnke, 2025). Slots' concise form allows more explicit object-relation modeling in visual reasoning (Ding et al., 2021). Disentangling objects facilitates compositional generation of future frames in video prediction (Villar-Corrales et al., 2023).

Mainstream OCL owes its success largely to Slot Attention (SA) (Locatello et al., 2020). It is essentially a form of iterative cross attention, where query slots compete to aggregate as much object information and discover objects as segmentation masks (Locatello et al., 2020). Such model is trained by minimizing reconstruction loss based on the aggregated slots, requiring no external supervision. Specifically, for image, query slots are usually sampled from some dataset-level Gaussian distributions (Jia et al., 2023) or initialized as some position priors (Kipf et al., 2022). Such queries contain little information about a specific sample, making iterative refinement necessary. For video, such aggrega-

[1]Department of Electrical Engineering and Automation, Aalto University, Espoo, Finland [2]Department of Computer Science, Aalto University, Espoo, Finland [3]Center for Machine Vision and Signal Analysis, University of Oulu, Oulu, Finland. Correspondence to: Rongzhen Zhao <rongzhen.zhao@aalto.fi>.

*Proceedings of the 43rd International Conference on Machine Learning*, Seoul, South Korea. PMLR 306, 2026. Copyright 2026 by the author(s).

tion transform proceeds recurrently across frames, where queries for the first frame are the same as in the image case while queries for non-first frames are transitioned from the previous frame's slots (Singh et al., 2022b). Unlike the first frame's queries, non-first frames' queries are already sample-specific.

As shown in Figure 1, to the best of our knowledge, all OCL methods based on SA and its variants confront these two issues. (*i1*) *Query cold-start* in SA iterations: For image or video's first frame, cold-start queries lack scene-specific information. The intrinsic cold-start queries hinder precise aggregation on image or video's first frame. (*i2*) *Transform homogeneity* in SA recurrences: For video frames, unlike the first frame, non-first frames' queries are much more informative, imposing different requirements on the aggregation transforms. The mainstream homogeneous transforms cannot adapt to such information gap.

Our solution *SmoothSA* is straightforward. (*s1*) Smoothing SA iterations on image or video's first frame: A tiny module *preheats* queries using input-feature information. It is trained by approximating current slots from queries, by self-distillation inside the OCL model. (*s2*) smoothing SA recurrences across video's first and non-first frames: Different aggregation transforms handle video's first and non-first frames respectively. It is realized by applying full iterations on the first frame while one on each non-first frame.

Briefly, our contributions are: (*c1*) for the first time addressing the query cold-start issue in SA iterations on image and video's first frame; (*c2*) for the first time addressing the transform homogeneity issue in SA recurrences across the first and non-first frames; (*c3*) new state-of-the-art on both image and video OCL benchmarks; (*c4*) consistent performance boosts on downstream advanced vision tasks.

## 2. Related Work

**Slot Attention on images and videos**. The seminal work Slot Attention (SA) (Locatello et al., 2020) proposes refining query slots with image features iteratively. Image OCL methods thereafter (Singh et al., 2022a; Seitzer et al., 2023; Wu et al., 2023b; Jiang et al., 2023; Kakogeorgiou et al., 2024; Zhao et al., 2025a;b;c;d) adopt such iterative design. The pioneering work STEVE (Singh et al., 2022b) extends SA to videos by conducting recurrent version of "image OCL" across frames, with queries transitioned from previous slots. Video OCL methods ever since (Kipf et al., 2022; Elsayed et al., 2022; Aydemir et al., 2023; Zadaianchuk et al., 2024; Manasyan et al., 2025; Li et al., 2025b;a; Zhao et al., 2026) adopt such recurrent design. Now that SA lies at the heart of mainstream OCL, the two issues described above are universal. We are the first to address them explicitly.

**Query initialization for Slot Attention iterations**. For images, queries are the starting point for aggregation via SA iterations. The contradiction is no sample-specific cues available before aggregation. SA (Locatello et al., 2020) initializes queries by drawing multiple samples from a global Gaussian distribution that fits and embeds global cues for a dataset. BO-QSA (Jia et al., 2023) learns multiple Gaussian distributions so that more distinct cues are captured, thus enabling better aggregation. However, queries are still cold-start. MetaSlot (Liu et al., 2025) firstly initializes queries from multiple Gaussians for draft aggregation, and then replaces the draft slots with object prototypes from a codebook (Van Den Oord et al., 2017) for re-initialized aggregation. None addressed the query cold-start issue explicitly.

**Query prediction for Slot Attention recurrences**. For video's first frame, the queries can be obtained in the same way as in the image case, or by transforming initial object bounding boxes as in SAVi (Kipf et al., 2022) and SAVi++ (Elsayed et al., 2022), at the cost of extra expensive annotations. For non-first frames, the queries are predicted from the previous frame's slots. STEVE (Singh et al., 2022b) and most other OCL methods use a Transformer encoder block for such recurrent prediction. STATM (Li et al., 2025b) and SlotPi (Li et al., 2025a) employ some auto-regressive Transformer encoder. RandSF.Q (Zhao et al., 2026), state-of-the-art, implicitly learns transition dynamics with random slot-feature pairs, obtaining significant performance boosts. None addressed the transform homogeneity issue explicitly.

## 3. Proposed Method

Mainstream image or video OCL methods confront two issues: query cold-start in SA iterations on image or video's first frame, and transform homogeneity in SA recurrences across video's first and non-first frames. Our *SmoothSA* preheats queries to smooth the iterations, and differentiate transforms to smooth the recurrences.

### 3.1. Slot Attention Iteration and Recurrence

Mainstream OCL adopts the encode-aggregate-decode model design (Zhao et al., 2025c). The encoder encodes image or video frames into features, the aggregator aggregates features into slots, and the decoder decodes slots into the reconstruction of the input in some form for self-supervision.

**SA iterations on image or video's first frame**

The aggregator $\phi_{\mathrm{a}}$, an SA module, takes input features $\boldsymbol{F}_1 \in \mathbb{R}^{h \times w \times c}$ as the key and value, and iteratively refines cold-start queries $\boldsymbol{Q}_1 \in \mathbb{R}^{n \times c}$ into object (and background) representations $\boldsymbol{S}_1 \in \mathbb{R}^{n \times c}$, i.e., slots, along with corresponding segmentation masks $\boldsymbol{M}_1 \in \mathbb{R}^{n \times h \times w}$:

$$\boldsymbol{Q}_1 = \phi_{\mathrm{n}}(\boldsymbol{C}) \qquad (1)$$

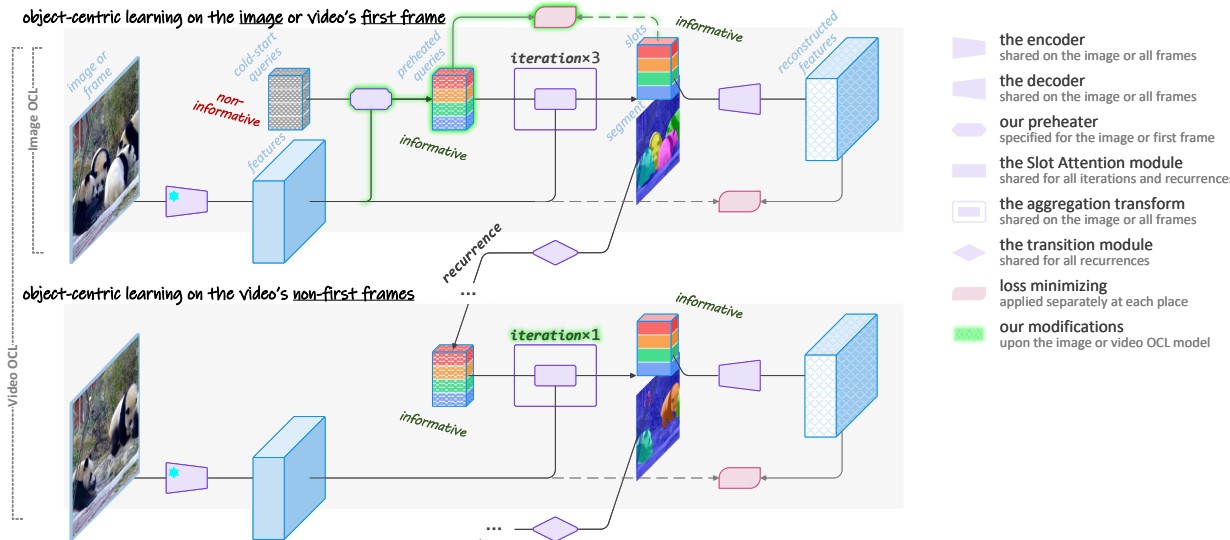

**Figure 2.** The overall model and our modifications. (*upper*) For image OCL, we preheat the cold-start queries to be informative so as to smooth SA iterations on image (or video's first frame). Our preheater is a tiny module trained to predict vectors approximating the slots as the preheated queries from the cold-start queries and image features. (*upper + lower*) For video OCL, we differentiate the homogeneous transforms to adapt to the first and non-first queries, non-informative and informative respectively, to smooth SA recurrences across all frames. This is achieved by using three SA iterations on the first frame and one on non-first frames.

$$\boldsymbol{S}_1, \boldsymbol{M}_1 = \boldsymbol{\Phi}_\mathrm{a}(\boldsymbol{Q}_1, \boldsymbol{F}_1) \qquad (2)$$

where the aggregation transform $\boldsymbol{\Phi}_\mathrm{a}$ can be expanded into:

$$\boldsymbol{S}_1^{(0)} := \boldsymbol{Q}_1 \qquad (2a)$$

$$\boldsymbol{S}_1^{(i)}, \boldsymbol{M}_1^{(i)} = \phi_\mathrm{a}(\boldsymbol{S}_1^{(i-1)}, \boldsymbol{F}_1) \quad i = 1, 2, 3 \qquad (2b)$$

$$\boldsymbol{S}_1, \boldsymbol{M}_1 := \boldsymbol{S}_1^{(3)}, \boldsymbol{M}_1^{(3)} \qquad (2c)$$

In Equation (1), if $\boldsymbol{C}$ is #slots $n$ to use, then $\phi_\mathrm{n}$ samples $n$ vectors as $\boldsymbol{Q}_1$ from its trainable Gaussian distribution(s) (Locatello et al., 2020; Jia et al., 2023); If $\boldsymbol{C}$ is object bounding boxes in video's first frame, then $\phi_\mathrm{n}$ projects $\boldsymbol{C}$ into $\boldsymbol{Q}_1$ (Kipf et al., 2022; Elsayed et al., 2022). Anyway, $\boldsymbol{Q}_1$ lacks sample-specific information, i.e., being cold-start.

---

**Intuition & Formalism 1**

As the aggregator iteratively refines non-informative queries into informative slots, it would work better if we can preheat queries to be closer to slots.

- - - - - - - - - - - - - - - - - - - - - - - - - - - - - -

Given $\boldsymbol{F}$, $\phi_\mathrm{a}$ is usually a *contraction* (Jia et al., 2023):

$$||\phi_\mathrm{a}^{(i)}(\boldsymbol{x}) - \phi_\mathrm{a}^{(i)}(\boldsymbol{y})|| \le \alpha^i ||\boldsymbol{x} - \boldsymbol{y}|| \quad \text{where } \alpha \in [0, 1) \quad (3)$$

By the *Banach fixed point* theorem, there is a unique fixed point $\boldsymbol{S}^* = \phi_\mathrm{a}(\boldsymbol{S}^*)$, and for every $\boldsymbol{x}$:

$$||\phi_\mathrm{a}^{(i)}(\boldsymbol{x}) - \boldsymbol{S}^*|| \le \alpha^i ||\boldsymbol{x} - \boldsymbol{S}^*|| \qquad (4)$$

Find some preheater $\phi_\mathrm{p}$ to make $\phi_\mathrm{p}(\boldsymbol{Q})$ closer to $\boldsymbol{S}^*$ than $\boldsymbol{Q}$:

$$||\phi_\mathrm{p}(\boldsymbol{Q}) - \boldsymbol{S}^*|| \le q||\boldsymbol{Q} - \boldsymbol{S}^*|| \quad \text{where } q \in [0, 1) \quad (5)$$

---

And then:

$$||\phi_\mathrm{a}^{(3)}(\phi_\mathrm{p}(\boldsymbol{Q})) - \boldsymbol{S}^*|| \le \alpha^3 ||\phi_\mathrm{p}(\boldsymbol{Q}) - \boldsymbol{S}^*|| \le q\alpha^3 ||\boldsymbol{Q} - \boldsymbol{S}^*|| \qquad (6)$$

Compared with not using preheater,

$$||\phi_\mathrm{a}^{(3)}(\boldsymbol{Q}) - \boldsymbol{S}^*|| \le \alpha^3 ||\boldsymbol{Q} - \boldsymbol{S}^*|| \qquad (7)$$

The preheated run is closer to the fixed point after three iterations than the non-preheated run. Refer to Section 3.2 for implementation and to Section 5 for visualized verification.

---

**SA recurrences across video's first and non-first frames**

The aggregator $\phi_\mathrm{a}$ is shared across all video frames recurrently. For the first frame, is identical to the image case formulated in Equations (1) and (2); while for non-first frames, queries $\boldsymbol{Q}_t$ are recurrently transitioned from previous frame's slots $\boldsymbol{S}_{t-1}$ by the transitioner $\phi_\mathrm{r}$:

$$\boldsymbol{Q}_t = \phi_\mathrm{r}(\boldsymbol{S}_{t-1}) \quad t \ge 2 \qquad (8)$$

$$\boldsymbol{S}_t, \boldsymbol{M}_t = \boldsymbol{\Phi}_\mathrm{a}'(\boldsymbol{Q}_t, \boldsymbol{F}_t) \qquad (9)$$

where the aggregation transform $\boldsymbol{\Phi}_\mathrm{a}'$ can be expanded into:

$$\boldsymbol{S}_t^{(0)} := \boldsymbol{Q}_t \qquad (9a)$$

$$\boldsymbol{S}_t^{(i)}, \boldsymbol{M}_t^{(i)} = \phi_\mathrm{a}(\boldsymbol{S}_t^{(i-1)}, \boldsymbol{F}_t) \quad i = 1, 2, 3 \qquad (9b)$$

$$\boldsymbol{S}_t, \boldsymbol{M}_t := \boldsymbol{S}_t^{(3)}, \boldsymbol{M}_t^{(3)} \qquad (9c)$$

In Equation (8), $\boldsymbol{S}_{t-1}$ is the informative representation of the previous frame and is used to predict $\boldsymbol{Q}_t$ by transition

dynamics (Singh et al., 2022b; Zhao et al., 2026), thus $\boldsymbol{Q}_t$ is already informative to current frame. In contrast, $\boldsymbol{Q}_1$ is non-informative. The identical or homogeneous transforms, i.e., $\boldsymbol{\Phi}'_a \equiv \boldsymbol{\Phi}_a$, cannot adapt to such an information gap.

---

**Intuition & Formalism 2**

As information in the first and non-first frames's queries are different while the aggregator is recurrently identically shared across frames, it would work better if we can adjust the number of iterations respectively.

- - - - - - - - - - - - - - - - - - - - - - - - - - - - -

We supplement the OCL decoding and supervision parts: Reconstruction is $\boldsymbol{X}' = \phi_d(\boldsymbol{S})$ and loss is $l = \mathrm{MSE}(\boldsymbol{X}', \boldsymbol{X})$. As $\phi_a$ is a *contraction*, according to *Lipschitz Jacobian bounds*,

$$\|\frac{\partial \phi_a(\boldsymbol{S})}{\partial \boldsymbol{S}}\| \leq \alpha \quad \text{(consistent with contraction)} \tag{10}$$

$$\|\frac{\partial \phi_a(\boldsymbol{S})}{\partial \boldsymbol{\theta}_a}\| \leq B \quad \text{(bound on param strength per iter)} \tag{11}$$

$$\|\frac{\partial \phi_d(\boldsymbol{S})}{\partial \boldsymbol{S}}\| \leq L \quad \text{(bound on the largest loss)} \tag{12}$$

Unroll the iterations. Let $\boldsymbol{J}_i := \frac{\partial \phi_a(\boldsymbol{S}^{(i-1)})}{\partial \boldsymbol{S}}$ and $\boldsymbol{U}_i := \frac{\partial \phi_a(\boldsymbol{S}^{(i-1)})}{\partial \boldsymbol{\theta}_a}$, $\boldsymbol{D} := \frac{\partial \phi_d(\boldsymbol{S})}{\partial \boldsymbol{S}}$ and $\boldsymbol{G}_{\boldsymbol{X}} := \frac{\partial l}{\partial \boldsymbol{X}'}$. And the derivative of $\boldsymbol{S}^{(3)}$ w.r.t $\boldsymbol{\theta}_a$ is the sum of contributions from each iteration:

$$\frac{\boldsymbol{S}^{(3)}}{\partial \theta_a} = \Sigma_{i=1}^{3}(\Pi_{m=i+1}^{3}\boldsymbol{J}_m)\boldsymbol{U}_i \tag{13}$$

By *chain rule*, the full gradient is

$$\frac{\partial l}{\partial \boldsymbol{\theta}_a} = \boldsymbol{G}_{\boldsymbol{X}}^{T}\boldsymbol{D}\frac{\partial \boldsymbol{S}^3}{\partial \boldsymbol{\theta}_a} = \boldsymbol{G}_{\boldsymbol{X}}^{T}\boldsymbol{D}\Sigma_{i=1}^{3}(\Pi_{m=i+1}^{3}\boldsymbol{J}_m)\boldsymbol{U}_i \tag{14}$$

For the frame with $i$ iterations,

$$\|\frac{\partial l}{\partial \phi_a}\| \leq \|\boldsymbol{G}_{\boldsymbol{X}}\|LB\Sigma_{i=0}^{i-1}\alpha^i = \|\boldsymbol{G}_{\boldsymbol{X}}\|LB\frac{1-\alpha^i}{1-\alpha}$$
$$< \|\boldsymbol{G}_{\boldsymbol{X}}\|LB\frac{1}{1-\alpha} \tag{15}$$

where on the right, only $\|\boldsymbol{G}_{\boldsymbol{X}}\|$ depends on #iterations $i$. In practice, for the first frame $\|\boldsymbol{G}_{\boldsymbol{X}}\|$ tends to be large and more iterations reduces it, while for non-first frames $\|\boldsymbol{G}_{\boldsymbol{X}}\|$ tends to be small and less iterations are needed. Refer to Section 3.3 for implementation and to Section 5 for visualized verification.

---

### 3.2. Preheating Cold-start Queries

To address the query cold-start issue and smooth SA iterations on image or video's first frame, we preheat queries with rich information from input features. A tiny module is trained via self-distillation inside the OCL model, to predict vectors approximating the aggregated slots as the preheated queries, from queries and conditioned on input features.

Firstly, we insert this between Equations (1) and (2):

$$\boldsymbol{Q}_1^* = \phi_p(\boldsymbol{Q}_1, \boldsymbol{F}_1) \tag{16}$$

where the preheater $\phi_p$ is parameterized as a single Transformer decoder block (Vaswani et al., 2017), whose self-attention and cross-attention are switched. This is because exchanging information among non-informative queries firstly is meaningless. Refer to Table 5 for why not using an extra SA module as the preheater, and for why switching the self-attention and cross-attention.

Secondly, we replace Equation (2a) with:

$$\boldsymbol{S}_1^{(0)} := \mathrm{sg}(\boldsymbol{Q}_1^*) \tag{17}$$

where $\mathrm{sg}(\cdot)$ is stopping gradient. Stopping gradient flow from the SA module $\phi_a$ to the preheated queries $\boldsymbol{Q}_1^*$ disentangles the training of $\phi_a$ and $\phi_p$. Refer to Table 5 for why stopping gradient flow on the preheated queries.

Lastly, we train our preheater $\phi_p$ by this objective:

$$\arg\min_{\boldsymbol{C},\phi_n,\phi_p} \mathrm{MSE}(\boldsymbol{Q}_1^*, \mathrm{sg}(\boldsymbol{S}_1)) \tag{18}$$

where the MSE loss is combined with the original OCL loss(es). To ensure sufficient training, we can use a relatively large coefficient on it. Refer to Table 5 for what loss weight to set for such preheating loss.

*Remark 1.* Our preheater is trained with OCL intermediate results as the ground-truth, without any external supervision, forming rigid self-distillation. This is also bootstrap, as good slots $\boldsymbol{S}_1$ leads to better preheated queries $\boldsymbol{Q}_1^*$, and in turn better $\boldsymbol{Q}_1^*$ leads to better $\boldsymbol{S}_1$.

*Remark 2.* Our preheater is similar to but lighter than the SA module, thus introducing negligible computation overhead inside the whole model. Refer to Table 6 for details.

### 3.3. Differentiating Homogeneous Transforms

To address the transform homogeneity issue and smooth SA recurrences across video's first and non-first frames, we differentiate transforms for the first and non-first frames respectively. To adapt to different transform requirements due to the information gap between the first and non-first queries, full and single SA iterations are used respectively.

As mentioned above, the first-frame transform $\boldsymbol{\Phi}_a$ and non-first frame transforms $\boldsymbol{\Phi}'_a$ are identical in mainstream methods. There are two ways to differentiate them: (1) use separate SA parameters for $\boldsymbol{\Phi}_a$ and $\boldsymbol{\Phi}'_a$; (2) use different number of iterations for $\boldsymbol{\Phi}_a$ and $\boldsymbol{\Phi}'_a$. We choose the second solution. This is because $\boldsymbol{\Phi}_a$ and $\boldsymbol{\Phi}'_a$ should learn the general aggregation capability in each SA iteration and sharing enforces this. Refer to Table 5 for numbers of iterations to set.

We empirically use one SA iterations in non-first frame transforms $\boldsymbol{\Phi}'_a$, while always three iterations in the first frame transform $\boldsymbol{\Phi}_a$. Namely, we keep Equations (2b) and (2c)

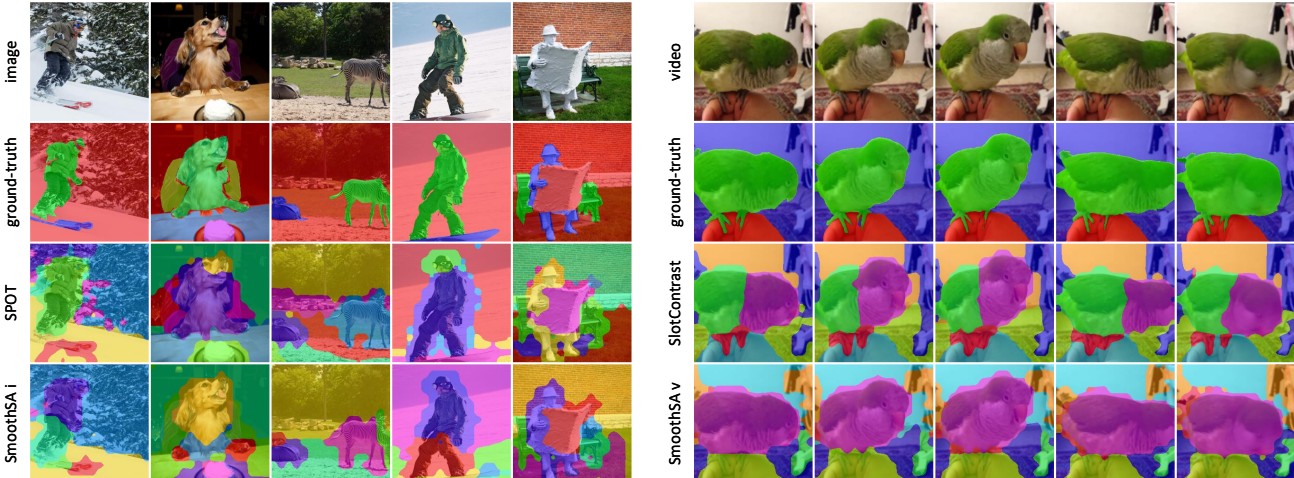

*Figure 3.* Qualitative results of our SmoothSA on images (*left*) and videos (*right*), compared with SotA methods SPOT and SlotContrast respectively.

| | ClevrTex #slot=11 | | | | COCO #slot=7 | | | | VOC #slot=6 | | | |
|---|---|---|---|---|---|---|---|---|---|---|---|---|
| | ARI | ARI$_{fg}$ | mBO | mIoU | ARI | ARI$_{fg}$ | mBO | mIoU | ARI | ARI$_{fg}$ | mBO | mIoU |
| SLATE | 17.4$_{\pm2.9}$ | 87.4$_{\pm1.7}$ | 44.5$_{\pm2.2}$ | 43.3$_{\pm2.4}$ | 17.5$_{\pm0.6}$ | 28.8$_{\pm0.3}$ | 26.8$_{\pm0.3}$ | 25.4$_{\pm0.3}$ | 18.6$_{\pm0.1}$ | 26.2$_{\pm0.5}$ | 37.2$_{\pm0.5}$ | 36.1$_{\pm0.4}$ |
| DINOSAUR | 50.7$_{\pm24.1}$ | 89.4$_{\pm0.3}$ | 53.3$_{\pm5.0}$ | 52.8$_{\pm5.2}$ | 18.2$_{\pm1.0}$ | 37.0$_{\pm1.2}$ | 28.3$_{\pm0.5}$ | 26.9$_{\pm0.5}$ | 21.5$_{\pm0.7}$ | 36.2$_{\pm1.3}$ | 40.6$_{\pm0.6}$ | 39.7$_{\pm0.6}$ |
| SlotDiffusion | 66.1$_{\pm1.3}$ | 82.7$_{\pm1.6}$ | 54.3$_{\pm0.5}$ | 53.4$_{\pm0.8}$ | 17.7$_{\pm0.5}$ | 29.0$_{\pm0.1}$ | 27.0$_{\pm0.4}$ | 25.6$_{\pm0.4}$ | 17.0$_{\pm1.2}$ | 21.7$_{\pm1.8}$ | 35.2$_{\pm0.9}$ | 34.0$_{\pm1.0}$ |
| SPOT | 25.6$_{\pm1.4}$ | 77.1$_{\pm0.5}$ | 48.2$_{\pm0.6}$ | 46.3$_{\pm0.7}$ | 23.7$_{\pm0.5}$ | 40.4$_{\pm0.5}$ | 30.9$_{\pm0.2}$ | 29.3$_{\pm0.2}$ | 24.5$_{\pm0.3}$ | 31.0$_{\pm0.8}$ | 40.1$_{\pm0.2}$ | 38.6$_{\pm0.3}$ |
| DIAS | 80.9$_{\pm0.3}$ | 79.1$_{\pm0.3}$ | 63.3$_{\pm0.1}$ | 61.9$_{\pm0.0}$ | 25.6$_{\pm0.1}$ | 41.2$_{\pm0.3}$ | 31.7$_{\pm0.1}$ | 30.2$_{\pm0.1}$ | 30.9$_{\pm0.5}$ | 33.5$_{\pm0.7}$ | 43.4$_{\pm0.5}$ | 42.4$_{\pm0.5}$ |
| SmoothSA$^i$ | 76.8$_{\pm0.7}$ | 82.2$_{\pm1.7}$ | 60.6$_{\pm0.4}$ | 58.9$_{\pm0.6}$ | 29.3$_{\pm1.0}$ | 41.3$_{\pm1.2}$ | 33.4$_{\pm0.2}$ | 31.8$_{\pm0.2}$ | 35.0$_{\pm0.5}$ | 33.6$_{\pm1.2}$ | 45.2$_{\pm0.3}$ | 43.9$_{\pm0.3}$ |

*Table 1.* Object discovery on images. Input resolution is 224×224; DINO2 ViT-S/14 is for encoding.

unchanged, while replacing Equations (9b) and (9c) with:

$$\boldsymbol{S}_t^{(i)}, \boldsymbol{M}_t^{(i)} = \phi_{\mathrm{a}}(\boldsymbol{S}_t^{(i-1)}, \boldsymbol{F}_t) \quad i = 1 \tag{19b}$$

$$\boldsymbol{S}_t, \boldsymbol{M}_t := \boldsymbol{S}_t^{(1)}, \boldsymbol{M}_t^{(1)} \tag{19c}$$

For conditional SA like in SAVi (Kipf et al., 2022) and SAVi++ (Elsayed et al., 2022), they use homogeneous aggregation transforms, consisting of one single SA iteration for all frames. But we still use three SA iterations on the first frame and one on non-first frames. They believe that objects' bounding boxes as query initialization is informative enough. But in fact, they still carry little object information, except the spatial information. Thus more iterations on the first frame is still necessary. Their ablation study leads them to believe that one iteration is better than more just because they were not aware of such recurrent transform homogeneity issue.

*Remark 3.* Our differentiated transforms have SA iterations in the recurrences, thus reducing computation overhead on videos. Refer to Table 6 for details.

## 4. Experiment

To evaluate our object representation quality, we conduct experiments on object discovery, object recognition and visual question answering, each using three random seeds.

### 4.1. Instantiating SmoothSA

As shown in Figure 2, our OCL model with SmoothSA is built on DIAS (Zhao et al., 2025d) for images and on RandSF.Q (Zhao et al., 2026) for videos, respectively. These two state-of-the-art (SotA) methods share most designs except techniques specific to image and video. For image OCL, we remove slots pruning tricks from DIAS, and then replace its SA variant with our SmoothSA. For video OCL, we use RandSF.Q as it is, and then replace its SA with our SmoothSA. Thus we have models SmoothSA$^i$ and SmoothSA$^v$, where $i$ is image and $v$ is video.

Note that for conditional video OCL like SAVi (Kipf et al., 2022) and SAVi++ (Elsayed et al., 2022), the authors always use one SA iteration on all frames. But whether it is conditional or not, we always use three SA iterations on the first frame while one iteration on non-first frames.

### 4.2. Object Discovery

In mainstream OCL methods, attention maps of the slots are binarized as object segmentation, i.e., discovering objects. This intuitively reflects slots' representation quality.

| | MOVi-C #slot=11, conditional | | | | MOVi-D #slot=21, conditional | | | | YTVIS-HQ #slot=7 | | | |
|---|---|---|---|---|---|---|---|---|---|---|---|---|
| | ARI | $ARI_{fg}$ | mBO | mIoU | ARI | $ARI_{fg}$ | mBO | mIoU | ARI | $ARI_{fg}$ | mBO | mIoU |
| STEVE | – | – | – | – | $17.5_{\pm0.6}$ | $28.8_{\pm0.3}$ | $26.8_{\pm0.3}$ | $25.4_{\pm0.3}$ | – | – | – | – |
| VideoSAUR | $41.9_{\pm1.1}$ | $53.3_{\pm2.1}$ | $16.1_{\pm0.4}$ | $14.8_{\pm0.4}$ | $22.5_{\pm5.0}$ | $40.0_{\pm20.1}$ | $11.6_{\pm6.6}$ | $10.8_{\pm6.1}$ | $33.8_{\pm0.7}$ | $49.2_{\pm0.5}$ | $29.9_{\pm0.4}$ | $29.7_{\pm0.4}$ |
| SlotContrast | $64.6_{\pm9.4}$ | $59.9_{\pm5.3}$ | $27.7_{\pm3.0}$ | $25.8_{\pm2.9}$ | $45.3_{\pm4.1}$ | $63.9_{\pm0.2}$ | $26.7_{\pm1.0}$ | $25.1_{\pm1.0}$ | $37.2_{\pm0.6}$ | $49.4_{\pm1.1}$ | $33.0_{\pm0.2}$ | $32.8_{\pm0.1}$ |
| RandSF.Q | $65.4_{\pm10.7}$ | $67.4_{\pm2.1}$ | $29.2_{\pm3.8}$ | $26.8_{\pm3.7}$ | $41.6_{\pm3.7}$ | $77.5_{\pm1.0}$ | $27.4_{\pm1.0}$ | $25.6_{\pm1.0}$ | $40.1_{\pm0.6}$ | $58.0_{\pm1.0}$ | $37.6_{\pm0.4}$ | $37.2_{\pm0.4}$ |
| SmoothSA[v] | $50.9_{\pm1.6}$ | $69.0_{\pm0.3}$ | $31.7_{\pm0.8}$ | $30.2_{\pm0.8}$ | $43.8_{\pm1.5}$ | $70.5_{\pm0.7}$ | $31.4_{\pm0.4}$ | $30.2_{\pm0.4}$ | $42.4_{\pm0.8}$ | $63.0_{\pm3.4}$ | $38.9_{\pm0.7}$ | $38.3_{\pm0.6}$ |
| update 20260425 below | | | | | | | | | | | | |
| | MOVi-C #slot=11, conditional | | | | MOVi-E #slot=24, conditional | | | | YTVIS-2022 #slot=7 | | | |
| | ARI | $ARI_{fg}$ | mBO | mIoU | ARI | $ARI_{fg}$ | mBO | mIoU | ARI | $ARI_{fg}$ | mBO | mIoU |
| VideoSAUR | $41.9_{\pm1.1}$ | $53.3_{\pm2.1}$ | $16.1_{\pm0.4}$ | $14.8_{\pm0.4}$ | $17.4_{\pm2.5}$ | $34.6_{\pm20.7}$ | $8.3_{\pm4.9}$ | $7.5_{\pm4.3}$ | $33.4_{\pm0.8}$ | $48.2_{\pm0.7}$ | $27.2_{\pm0.3}$ | $26.8_{\pm0.3}$ |
| SlotContrast | $64.6_{\pm9.4}$ | $59.9_{\pm5.3}$ | $27.7_{\pm3.0}$ | $25.8_{\pm2.9}$ | $29.9_{\pm4.9}$ | $70.6_{\pm3.8}$ | $20.7_{\pm1.4}$ | $19.3_{\pm1.2}$ | $35.2_{\pm0.8}$ | $51.4_{\pm0.7}$ | $29.7_{\pm0.5}$ | $29.3_{\pm0.6}$ |
| RandSF.Q | $65.4_{\pm10.7}$ | $67.4_{\pm2.1}$ | $29.2_{\pm3.8}$ | $26.8_{\pm3.7}$ | $30.5_{\pm1.2}$ | $82.1_{\pm3.1}$ | $23.0_{\pm1.2}$ | $21.6_{\pm1.4}$ | $37.9_{\pm1.3}$ | $51.8_{\pm1.2}$ | $32.2_{\pm1.8}$ | $31.5_{\pm1.8}$ |
| SmoothSA[v] | $50.9_{\pm1.6}$ | $69.0_{\pm0.3}$ | $31.7_{\pm0.8}$ | $30.2_{\pm0.8}$ | $36.7_{\pm0.6}$ | $73.6_{\pm0.6}$ | $28.6_{\pm0.1}$ | $27.4_{\pm0.1}$ | $42.0_{\pm0.6}$ | $59.0_{\pm2.1}$ | $36.0_{\pm0.5}$ | $34.9_{\pm0.6}$ |

*Table 2.* Object discovery on videos. Input resolution is 224×224; DINO2 ViT-S/14 is for encoding.

On image datasets ClevrTex[1], COCO[2] and VOC[3], we compare our SmoothSA[i] with baselines SLATE (Singh et al., 2022a), DINOSAUR (Seitzer et al., 2023), SlotDiffusion (Wu et al., 2023b), SPOT (Kakogeorgiou et al., 2024) (no distillation and finetuning tricks) and DIAS (Zhao et al., 2025d) (no slot pruning). On video dataset YouTube Video Instance Segmentation[4] (YTVIS) the high-quality version[5], we compare our SmoothSA[v] with baselines STEVE (Singh et al., 2022b), VideoSAUR (Zadaianchuk et al., 2024), SlotContrast (Manasyan et al., 2025) and RandSF.Q (Zhao et al., 2026).

The performance metrics are ARI[6], $ARI_{fg}$ (foreground), mBO (Uijlings et al., 2013) and mIoU[7]. ARI score is calculated with the segmentation area as the weight, thus ARI mainly reflects how well the background is segmented while $ARI_{fg}$ reflects how well large objects are segmented. mBO shows how objects that are best overlapped with the ground-truth are segmented. mIoU is the most strick metric. Note that, unless otherwise specified, we use image ARI, $ARI_{fg}$, mBO and mIoU for object discovery on images, while using video ones for object discovery on videos.

As shown in Table 1, on synthetic dataset ClevrTex, our SmoothSA[i] is as competitive as the latest SotA DIAS and significantly better than former SotA SPOT in all metrics. On real-world dataset COCO and VOC, our SmoothSA[i] is consistently better than DIAS in all metrics. Our method achieves overall new SotA in ARI, mBO and mIoU, except relative limited performance boosts in $ARI_{fg}$.

As shown in Table 2, on real-world video dataset YTVIS, our SmoothSA[v] defeats all baselines by a large margin, even including the latest SotA method RandSF.Q, which has already pushed the older SotA performance significantly forward by up to 10 points. On synthetic video datasets MOVi-C/D, our method demonstrates overall advantages, especially when measured by the most strick metric, mIoU.

### 4.3. Object Recognition

Besides the byproduct segmentation, recognizing the discovered objects' attributes like class and bounding box from the slots can directly reflect the representation quality.

On real-world image dataset COCO, we compare our SmoothSA[i] with baseline SPOT (Kakogeorgiou et al., 2024). On real-world video dataset YTVIS, we compare our SmoothSA[v] with baseline SlotContrast (Manasyan et al., 2025). We follow the routine of (Seitzer et al., 2023): firstly convert all images into slots representation, with some threshold filtering; then train a two-layer MLP model to classify and regress the matched object's class label and bounding box coordinates in a supervised way. We use top1 and top3 accuracy to measure the classification performance, and box IoU to measure the regression performance.

| | class top1 | top3 | bbox IoU | #match |
|---|---|---|---|---|
| | COCO #slot=7 | | | |
| SPOT +MLP | $88.7_{\pm0.3}$ | $97.2_{\pm0.1}$ | $48.6_{\pm0.1}$ | $5061_{\pm30}$ |
| SmoothSA[i] +MLP | $89.3_{\pm0.5}$ | $97.3_{\pm0.2}$ | $49.5_{\pm1.1}$ | $5210_{\pm223}$ |
| | YTVIS-HQ #slot=7 | | | |
| SlotContrast+MLP | $85.8_{\pm0.3}$ | $95.8_{\pm0.4}$ | $51.5_{\pm0.5}$ | $9249_{\pm41}$ |
| SmoothSA[v] +MLP | $90.4_{\pm0.2}$ | $97.6_{\pm0.1}$ | $42.6_{\pm1.4}$ | $8957_{\pm34}$ |
| update 20260425 below | | | | |
| | class top1 | top3 | bbox IoU | #match |
| | YTVIS-2022 #slot=7 | | | |
| SlotContrast+MLP | $87.1_{\pm0.2}$ | $96.4_{\pm0.1}$ | $48.2_{\pm0.3}$ | $19943_{\pm156}$ |
| SmoothSA[v] +MLP | $91.0_{\pm0.2}$ | $97.6_{\pm0.0}$ | $39.2_{\pm1.0}$ | $18864_{\pm200}$ |

*Table 3.* Object recognition on images and videos.

As shown in Table 3, the object recognition accuracy on

---

[1] https://www.robots.ox.ac.uk/~vgg/data/clevrtex

[2] https://cocodataset.org

[3] http://host.robots.ox.ac.uk/pascal/VOC

[4] https://youtube-vos.org/dataset/vis

[5] https://github.com/SysCV/vmt?tab=readme-ov-file#hq-ytvis-high-quality-video-instance-segmentation-dataset

[6] https://scikit-learn.org/stable/modules/generated/sklearn.metrics.adjusted_rand_score.html

[7] https://scikit-learn.org/stable/modules/generated/sklearn.metrics.jaccard_score.html

both real-world complex images and videos are improved a lot by using our method as the slots representation extractor, compared with that using baseline methods. This demonstrates the high quality of our slots representation.

## 4.4. Visual Question Answering

In visual question answering (VQA) tasks, the visual modality slots are combined with language modality words embeddings, testing the representation versatility further.

For VQA on images, we compare our SmoothSA$^i$ plus multi-modal reasoning model Aloe (Ding et al., 2021) with baseline SPOT plus Aloe on real-world complex image dataset GQA[8]. For VQA on videos, we compare our SmoothSA$^v$ plus Aloe with baseline SlotContrast plus Aloe on synthetic video dataset CLEVRER[9]. Please note that for the image dataset, we use Aloe as it is while on the video dataset we introduce temporal embedding scheme from (Wu et al., 2023a). For the upstream OCL models, we firstly pre-train them on corresponding datasets and freeze them to represent samples as slots. These visual input along with textual inputs representing questions are fed into the Aloe model together, appended with a classification token. The output is obtained by projecting the transformed classification token into logits of all possible class labels, i.e., answers.

| | GQA #slot=7 | |
|---|---|---|
| | accuracy % | |
| SPOT | + Aloe | $52.3_{\pm 2.8}$ |
| SmoothSA$^i$ | + Aloe | $56.7_{\pm 1.9}$ |
| | CLEVRER #slot=7 | |
| | per option % | per question % |
| SlotContrast | + Aloe | $97.2_{\pm 1.1}$ | $95.6_{\pm 0.9}$ |
| SmoothSA$^v$ | + Aloe | $98.7_{\pm 0.4}$ | $96.9_{\pm 0.6}$ |

*Table 4.* Visual question answering on images and videos.

As shown in Table 4, using our method as the upstream model improves image VQA performance on dataset GQA by 4+ points. As for video VQA on CLEVRER, using our method as the upstream boosts the performance too, whether measured by per option or per question accuracy.

## 4.5. Ablation

We conduct ablation studies as shown in Table 5.

(*a*) **Query preheating related**: (*a.1*) Implementing our preheater as a Transformer decoder block is better than as a Slot Attention module; (*a.1.1*) Using a Transformer decoder block as preheater, switching its self- and cross-attention in it is beneficial; (*a.2*) Stop-gradient on preheated queries is better than not; (*a.3*) Setting preheating loss weight to 100 is better than other values;

---

[8]https://cs.stanford.edu/people/dorarad/gqa

[9]http://clevrer.csail.mit.edu

| ARI + ARI$_{fg}$ | |
|---|---|
| Preheater implementation @COCO | |
| a Transformer decoder block | $68.3_{\pm 0.8}$ |
| a Slot Attention module | $63.3_{\pm 1.4}$ |
| no preheater and preheat loss | $56.9_{\pm 2.5}$ |
| Switch cross-attention and self-attention in preheater @COCO | |
| Yes | $68.3_{\pm 0.8}$ |
| No | $49.6_{\pm 9.4}$ |
| Stop gradient on preheated query @COCO | |
| Yes | $68.3_{\pm 0.8}$ |
| No | $67.5_{\pm 2.9}$ |
| Preheating loss weight @COCO | |
| 10 | $59.7_{\pm 1.0}$ |
| 50 | $65.5_{\pm 0.4}$ |
| 100 | $68.3_{\pm 0.8}$ |
| 200 | $67.4_{\pm 1.3}$ |
| Use separate weights for first and non-first transforms @YTVIS | |
| separate | $52.3_{\pm 0.7}$ |
| shared | $68.3_{\pm 0.8}$ |
| Unconditional video OCL: first and non-first SA #iter @YTVIS | |
| 3+1 | $105.6_{\pm 2.2}$ |
| 1+1 | $97.4_{\pm 11.4}$ |
| 3+3 | $103.4_{\pm 6.8}$ |
| Conditional video OCL: first and non-first SA #iter @MOVi-C | |
| 3+1 | $136.3_{\pm 7.1}$ |
| 1+1 | $133.9_{\pm 15.0}$ |
| 3+3 | $132.7_{\pm 8.4}$ |

*Table 5.* Ablation studies.

| V100 | time / hours | memory / GB |
|---|---|---|
| SPOT @COCO, bs32 | 4.7 | 8.5 |
| DIAS @COCO, bs32 | 4.5 | 9.4 |
| SmoothSA @COCO, bs32 | 4.2 | 8.7 |
| SlotContrast @YTVIS, bs8 | 7.4 | 6.4 |
| RandSF.Q @YTVIS, bs8 | 6.8 | 5.2 |
| SmoothSA @YTVIS, bs8 | 7.1 | 5.2 |

*Table 6.* Computation overhead.

(*\**) *Additional intuition on the attention switch of our preheater.* (*i*) Information Vacuum: Initial queries are sampled from a global Gaussian distribution, i.e., image-agnostic noise. (*ii*) In standard SA → CA: Performing SA first forces non-informative queries to exchange information with each other – The (non-linear) combination of noises is still noise. (*iii*) In the switched order (CA → SA): Performing CA first anchors non-informative queries to the input features, i.e., preheating each query with image-specific distinct object cues. Only then the subsequent SA is meaningful.

(*b*) **Transform differentiating related**: (*b.1*) Using shared module weights on first-frame transform $\Phi_a$ and non-first-frame transforms $\Phi'_a$ is better than using separate weights; (*b.2*) For conditioned video OCL, using iteration numbers of 3 and 1 on first and non-first frames respectively is better than other combinations; (*b.3*) For unconditioned video OCL, using iteration numbers of 3 and 1 on first and non-first frames respectively is better than other combinations.

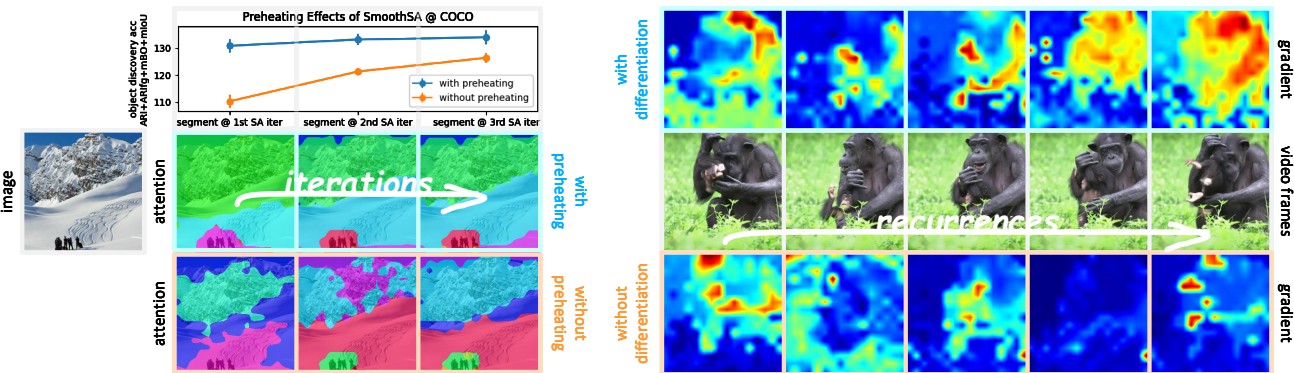

*Figure 4.* (*left*, *middle row*) Query preheating: good segmentation can be obtained at the initial SA iteration, with better segmentation at the end. (*right*, *top row*) Transform differentiation: balanced gradient signals can be obtained across SA recurrences, showing object contours.

## 5. Discussion

### Query preheating smooths SA iterations

The segmentation accuracies generally increase along with the SA iterations, so we expect that our query preheating provides better initial queries and the accuracies increase faster. Refer to Section 3.1 for formal analyses. To statistically analyze this, we training our SmoothSA$^i$ with and without query preheating on COCO, and count their respective object discovery/segmentation accuracy at each of the three SA iterations. In practice, those metrics (ARI, ARI$_{fg}$, mBO and mIoU), mostly show cluttered tendencies in iterations, thus we sum them as a more readable metric.

As shown in Figure 4 (*left*) the top row, on the whole dataset, segmentation accuracies with or without query preheating at three SA iterations increase steadily. But using query preheating obviously speeds it up and leads to better final accuracy than not. As shown in Figure 4 (*left*) the lower two rows, using preheating obtains good segmentation at the very beginning SA iteration, while not using preheating struggles with it in the first two SA iterations. Thus our query preheating really smooths SA iterations on the image. And it should be the same on the video's first frame.

### Transform differentiation smooths SA recurrences

The non-informative and informative queries of the first and non-first video frames generally require different transform capabilities through the SA recurrences, so we expect our transform differentiation provides better gradient signals during training. Refer to Section 3.1 for formal analyses. To statistically analyze this, we should count the per-frame gradients of the SA module, contributed by per-frame decoding. But in practice, such gradients are always merged together by mainstream deep learning libraries like PyTorch. Thus we take the per-frame gradients of per-frame input features as an indirect reflection. The gradient map is calculated by averaging the per-frame gradient absolute along the channel dimension, leaving spatial dimensions for visualization.

As shown in Figure 4 (*right*), we visualize the gradient maps with and without our transform differentiation given a video sample. The input features of both first and non-first frames receive more balanced gradient signals if using transform differentiation than not. Specifically, the gradient maps show better object contours and overall amplitudes with transform differentiation, while showing very unclear object contours and fluctuated amplitudes without it.

## 6. Conclusion

In this work, we propose a novel method SmoothSA, which addresses the query cold-start issue in SA iterations on image or video's first frame, and transform homogeneity issue in SA recurrences across video's first and non-first frames. We introduce two techniques, query preheating and transform differentiating, to address these two issues. Our SmoothSA achieves new state-of-the-art performance on object discovery, and also benefits downstream tasks including object recognition and visual question answering.

**Limitations and future works**. Intuitively our method has two possible limitations. For query preheating, if the aggregated slots are bad then the preheated queries are bad and in turn the slots can be even worse; For transform differentiating, we empirically use three and one iterations, instead of automatically, which may not fit all cases. Determining when these two limitations are prominent and how to overcome them are left for future works.

## Acknowledgment

We acknowledge the support of Finnish Center for Artificial Intelligence (FCAI), Research Council of Finland flagship program. We thank the Research Council of Finland for funding the projects grant no. 357301, 362407, 352788,

373780, 372999 and 353138. We also appreciate CSC - IT Center for Science, Finland, for granting access to supercomputers Mahti and Puhti, as well as LUMI, owned by the European High Performance Computing Joint Undertaking (EuroHPC JU) and hosted by CSC Finland in collaboration with the LUMI consortium. Furthermore, we acknowledge the computational resources provided by the Aalto Science-IT project through the Triton cluster.

## Impact Statement

This paper presents work whose goal is to advance the field of Machine Learning. There are many potential societal consequences of our work, none which we feel must be specifically highlighted here.

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

## A. Self-distillation Sensitivity to Poor Slot Representations

Our designs ensure the robustness. Architectural Isolation, as shown in Equation (17): we stop gradient flow between preheated queries and slots. This "firewall" disentangles our preheater from the SA module. Feature-Anchored Correction: Over SA contractive iterations, any query errors are exponentially attenuated (Equation 6) and the final representation is anchored by features (Banach fixed-point theorem). Even in the "worst case", our preheated queries provide a sample-specific prior, which is statistically superior to image-agnostic cold-start queries. Stress Test: We heavily corrupt the slots to test the stability: $S \equiv S(1 - \epsilon) + \mathcal{N}(0, 1)\epsilon$, where $\epsilon = 0.999$. We performed this 1k times evenly across 100k training steps. The negligible performance deltas verifies the robustness.

| @COCO (seed=42/43/44) | $MSE_{recon}$ | $MSE_{preheat}$ | ARI | $ARI_{fg}$ | mBO | mIoU |
|---|---|---|---|---|---|---|
| SmoothSA$^i$ | 0.329±0.08 | 6.9±1.5 | 29.3±1.0 | 41.3±1.2 | 33.4±0.2 | 31.8±0.2 |
| + random corruption | 0.304±0.13 | 7.0±1.4 | 29.1±1.2 | 41.8±1.7 | 32.8±0.8 | 31.2±0.5 |

## B. Validity of the Contraction Assumption

The Banach fixed-point interpretation in Section 3.1 was an illustrative theoretical motivation rather than a rigorous global guarantee. It provides the intuition for why smoothing optimization trajectories helps. In the context of OCL, Slot Attention is specifically designed to perform iterative refinement toward a stable grouping, which inherently seeks a local fixed point. Empirical Validation of Local Contraction: Following your constructive feedback, we empirically validate the local contraction behavior of SA. As shown below, we tracked the distance between slot representations across iterations, i.e., $d = \frac{\|S^{(i)} - S^{(i-1)}\|_2}{\|S^{(i-1)}\|_2}$. The distance always decays along the iterations and stabilizes, even at the beginning of training. And our preheater does speed up such contraction.

| distances between iters 1→2→3 @COCO | step=100/100000 | step=30000 | step=60000 |
|---|---|---|---|
| SmoothSA$^i$ | 0.57→0.45→0.39 | 1.01→0.56→0.25 | 0.74→0.31→0.16 |
| without preheat | 0.62→0.54→0.44 | 1.02→0.81→0.27 | 0.79→0.40→0.18 |

