# OpenReview forum: "Smoothing Slot Attention Iterations and Recurrences"
_ICML.cc/2026/Conference — ICML 2026 regular_

### Official Review · Reviewer_zoK4 · 2026-02-23

**Soundness:** 3
**Presentation:** 4
**Significance:** 3
**Originality:** 3
**Overall Recommendation:** 4
**Confidence:** 4

**Summary:**

The paper revisits the optimization dynamics of Slot Attention for object-centric learning and argues that unstable query initialization and inefficient iterative refinement limit representation quality and training efficiency. To address this issue, the authors propose a smoothing-based framework that introduces a learned query preheating mechanism, which initializes slots closer to their refined states, along with an iteration scheduling strategy that reduces refinement steps for subsequent frames. The method is motivated by viewing slot updates through a fixed-point optimization perspective, suggesting that improved initialization can accelerate convergence and stabilize learning. Empirical result suggests that the approach demonstrates improved training stability and competitive performance across object discovery and video-based object-centric learning benchmarks, indicating that better optimization dynamics can enhance slot-based representation learning.

**Compliance With Llm Reviewing Policy:**

Affirmed.

**Final Justification:**

My main concerns were addressed. The weaknesses I have mentioned have been clarified, and thus, I maintain my original score.

**Key Questions For Authors:**

(1) Validity of the contraction assumption. The theoretical motivation relies on interpreting the Slot Attention update as a contraction mapping. Could the authors provide empirical or analytical evidence supporting this assumption during training (e.g., convergence-rate measurements or sensitivity analyses with respect to initialization)? Without such validation, it remains unclear whether the proposed theory genuinely explains the observed improvements or primarily serves as an intuitive justification.

(2) Mechanism of improvement: Representation or Optimization? Do the authors have evidence that SmoothSA improves the quality or identifiability of learned object representations beyond stabilizing optimization dynamics? Analyses such as slot consistency across frames, semantic alignment, or disentanglement metrics would help clarify whether the gains arise from improved object-centric representations or mainly from more stable training.

(3) Robustness and generality of the preheating strategy. How robust is the proposed preheating mechanism to early training noise or imperfect slot assignments? In particular, is there a risk that preheating reinforces incorrect slot configurations learned at early stages?

**Limitations:**

The contraction assumption forms a central part of the paper’s theoretical motivation but remains largely unverified. In practice, attention-based updates are unlikely to satisfy contraction properties globally, raising concerns about whether the proposed mathematical framing genuinely explains the empirical behavior. Nevertheless, despite the lack of formal rigor, the paper makes a valuable attempt to connect slot refinement dynamics with iterative optimization theory. Interpreting improved initialization as approximating a fixed-point trajectory offers an appealing conceptual perspective that unifies query preheating, reduced refinement iterations, and temporal smoothing under a common framework.

More broadly, while the evaluation emphasizes segmentation metrics and downstream task performance, the paper provides limited analysis of the learned representation structure itself. Without examining whether slots become more semantically consistent, identifiable, or disentangled, it remains unclear whether SmoothSA advances object-centric learning at a conceptual level or primarily improves optimization stability and training efficiency.

**Strengths And Weaknesses:**

$\textbf{Strengths}$

(1) Clear identification of the structural weakness in Slot Attention. The paper explores a core challenge in OCL: the mismatch between cold-start queries and informative recurrent slots. Rather than proposing another architectural variant of Slot Attention, the authors diagnose a dynamical inconsistency in how Slot Attention operates across iterations and temporal recurrences. This framing is conceptually strong because it targets optimization behavior rather than only representation design, which is a relatively unexplored angle in OCL literature.

(2) Mechanistically motivated design. The introduction of query preheating is grounded in a contraction mapping perspective, arguing that initialization closer to the fixed point accelerates convergence. While simplified, the formulation connects Slot Attention updates to iterative optimization theory, offering a rare attempt to interpret slot refinement through dynamical systems reasoning rather than heuristic intuition. The work provided me with a stronger theoretical narrative compared to many OCL extensions.

(3) Significance and novelty. The paper does not introduce yet another architectural variant, but instead targets a fundamental optimization issue in Slot Attention: unstable initialization and inefficient iterative refinement. This diagnosis is meaningful because many recent OCL methods implicitly struggle with the same instability but treat it as an engineering issue.


$\textbf{Weaknesses}$

(1) A central theme of the manuscript is the argument that many failures in slot-based learning arise from optimization instability rather than representational limitations, which is an insightful and well-motivated perspective. Accordingly, the proposed method addresses both intra-frame refinement (iteration dynamics) and inter-frame recurrence through a unified principle of smoothing optimization trajectories. However, the theoretical justification relies on a strong contraction assumption that is neither empirically validated nor formally justified for attention-based architectures. In practice, attention modules are highly nonlinear and parameter-dependent, making it unlikely that global contraction properties hold; as a result, the Banach fixed-point interpretation appears largely illustrative rather than rigorous, even if it provides useful intuition.

(2) The preheater is trained to approximate slots produced by the same model, creating a bootstrap loop where errors in early slot formation may be reinforced. From an optimization perspective, this resembles target leakage of self-confirmation rather than principled supervision. The paper acknowledges instability only briefly but does not analyze failure modes such as error amplification or collapse.

(3) Reducing iterations for later frames and injecting feature-conditioned initialization changes the gradient flow and effective training dynamics. The performance improvements may stem from easier optimization toward temporal smoothness rather than genuinely better object decomposition. The paper does not disentangle whether representation quality improves or whether learning simply becomes more stable.

---

> ### Author Rebuttal · Authors · 2026-03-26
>
> # `W1` `Q1` `L1` Validity of the Contraction Assumption
> Thank you for the insightful comment regarding the non-linearity of attention.
> - **Clarification of our Intent**: The Banach fixed-point interpretation was an illustrative theoretical motivation rather than a rigorous global guarantee. It provides the intuition for why smoothing optimization trajectories helps.
> - **In the context of OCL**, Slot Attention is specifically designed to perform iterative refinement toward a stable grouping, which inherently seeks a local fixed point.
> - **Empirical Validation of Local Contraction**: Following your constructive feedback, we empirically validate the local contraction behavior of SA. As shown below, we tracked the distance between slot representations across iterations. The distance always decays along the iterations and stabilizes, even at the beginning of training. And our preheater does speed up such contraction.
>
> $distance= \frac { ||S^{(i)}-S^{(i-1)}||_2 } { ||S^{(i-1)}||_2 }$
>
> |  | distances between iters 1→2→3 @step=100/100k | distances between iters 1→2→3 @step=10000/100k |
> |:---:|:---:|:---:|
> | SmoothSA @COCO | 0.874→0.527→0.462 | 0.274→0.102→0.084 |
> | removing preheater | 0.881→0.594→0.587 | 0.629→0.287→0.191 |
>
> # `W2` `Q3` Bootstrap Sensitivity to Poor Slot Representations
> The reviewer correctly identifies a potential risk. However, our designs ensure the robustness:
> - **Architectural Isolation** (`Equation 17`): We stop gradient flow between preheated queries and slots. This "firewall" disentangles our preheater from the SA module.
> - **Feature-Anchored Correction** (`Lines 184-219`): Over the SA contractive iterations, any query errors are exponentially attenuated (`Equation 6`) and the final representation is anchored by the visual feature (Banach fixed-point theorem). Even in the "worst case", preheated queries provide a sample-specific prior, which is statistically superior to image-agnostic cold-start queries.
> - **Stress Test**: We heavily corrupt the slots to test the system's stability: $S = S \cdot (1-\epsilon) + \mathcal{N}(0,1) \cdot \epsilon$, where $\epsilon=0.999$. We performed this 1k times evenly across 100k training steps. The negligible performance deltas verifies the robustness.
>
> | @COCO (seed=42/43/44) | MSE | ARI | ARIfg | mBO | mIoU |
> |:---:|:---:|:---:|:---:|:---:|:---:|
> | SmoothSA | 0.1350±0.0005 | 29.3±1.0  | 41.3±1.2  | 33.4±0.2  | 31.8±0.2 |
> | + random corruption | 0.1356±0.0009 | 29.6±2.5  | 41.2±1.8  | 33.0±0.7  | 31.2±0.4 |
>
> # `W3` `Q2` `L2` Optimization Mechanism & Representation Quality
> Thanks for highlighting the need to distinguish genuine representation quality from optimization stabilization.
> - **Gradient Dynamics** (trunc_bp): We employ "truncated back-propagation" (trunc_bp) [BO-QSA], where ***only the last iteration propagates gradients***, regardless of the total #iters applied to a frame. Thus, our iter reducing for non-first frames does ***not alter gradient dynamics***. We will make this clearer in the revision.
>   - Besides, ablating trunc_bp shows SmoothSA remains competitive, confirming our gains stem from the architectural design rather than trunc_bp.
>
> | @YTVIS (seed=42/43/44) | ARI | ARIfg | mBO | mIoU |
> |:---:|:---:|:---:|:---:|:---:|
> | SmoothSA | 42.4±0.8 | 63.0±3.4 | 38.9±0.7 | 38.3±0.6 |
> | removing trunc_bp | 41.8±1.0 | 62.7±2.5 | 38.7±0.9 | 38.0±0.4 |
>
> - **Representation Quality & Disentanglement**: For semantic alignment and disentanglement, our downstream tasks in `Section 4.3` (Object Recognition) and `Section 4.4` (Visual Question Answering) explicitly validate it. We use the frozen learned slots to regress bounding boxes, classify semantics, and answer multi-modal questions. The significant and consistent performance boosts prove that our method improves object identifiability and semantic disentanglement, far beyond stabilizing optimization.
>   - Although we hope our `Sections 4.3-4.4` sufficiently addressed your concern, ***we are willing to provide further evidence under your further guidance if necessary.***
>
> # `*` Invitation for Further Discussion
> We welcome any follow-up questions and are eager to provide further evidence or clarification for your remaining concerns.
>
> ---
> [BO-QSA] Improving Unsupervised Object-centric Learning with Query Optimization

---

> > ### Author Rebuttal · Reviewer_zoK4 · 2026-04-01
> >
> > Thank you for the thorough and data-driven response. The additional empirical evidence and technical clarifications provided in the rebuttal effectively address my primary concerns regarding the SmoothSA framework. While I am satisfied with the core rebuttal, I have a few remaining questions to help clarify the method.
> >
> > Q1. Since the preheater effectively pre-contracts the queries, could the distance between $Q_{preheat}$ and $S_{t-1}$ be used to adaptively skip iterations entirely for very stable frames, or is the fixed schedule necessary for training stability?
> >
> > Q2. The stress test proved short-term robustness, but in extremely long video sequences with significant occlusion, is there any evidence of manifold drift where the preheated queries might slowly diverge from the true object distribution?
> >
> > Q3. Given that the iteration reduction relies on informative recurrences, is the final sequence quality heavily sensitive to the cold-start decomposition of the very first frame? Specifically, can a poor initial frame decomposition lead to a cascading failure that the preheater reinforces in subsequent frames?
> >
> > Overall, the rebuttal has significantly strengthened the paper’s foundation. I appreciate the authors' effort in providing new experiments.

---

> > > ### Author Response · Authors · 2026-04-02
> > >
> > > Thank you for your constructive engagement and for acknowledging the improvements in the rebuttal. We deeply appreciate your follow-up questions, which touch on critical aspects of recurrent optimization. Below, we provide clarifications and additional empirical evidence.
> > >
> > >
> > > # Response to Q1: Adaptive iteration skipping for stable frames
> > >
> > > This is an insightful suggestion.
> > >
> > > We must gently clarify that in our current architecture, the preheater is exclusively applied to the first frame (or in image OCL) to solve the extreme cold-start problem.
> > >
> > > During our development, we actually had experimented with applying the preheater to non-first frames. But it consistently degraded performance. We hypothesize this is because the transitioned queries ($S_{t-1}$) already contain strong, temporally smooth priors; injecting feature-conditioned preheating at every step disrupts this temporal consistency.
> > >
> > > Because preheating is restricted to the first frame, we cannot currently use a $Q_{\mathrm{preheat}}$ vs $S_{t-1}$ distance metric to skip iterations dynamically. However, exploring alternative stability metrics to adaptively skip iterations in later frames is a brilliant direction for future work on efficient video OCL.
> > >
> > >
> > >
> > > # Response to Q2: Manifold drift in long sequences with severe occlusion
> > >
> > > This is a vital point regarding long-term recurrent stability. Following the clarification in Q1, the preheater does not actively inject noise or cause manifold drift in subsequent frames, as it is only applied to the very first frame. Any potential drift would stem from the recurrent transition dynamics themselves.
> > >
> > > To explicitly test SmoothSA's robustness against drift in exceptionally long and highly occluded sequences, we still evaluated our model on the [OVIS].
> > > Our method still demonstrates superiority over the baseline.
> > >
> > > |     @OVIS    |    ARI   |   ARIfg  |    mBO   |   mIoU   |
> > > |:------------:|:--------:|:--------:|:--------:|:--------:|
> > > | SlotContrast | 37.2±1.0 | 44.8±1.0 | 20.1±0.8 | 17.1±0.9 |
> > > |   SmoothSA   | 39.3±2.3 | 47.5±0.4 | 21.0±0.4 | 19.5±0.4 |
> > >
> > > Please note:
> > > - The segmentation annotation for the val/test subsets are not available, thus we train on the val/test subsets while evaluating on the train subset;
> > > - The official evaluation server uses metric ***AP***, which requires confidence-thresholdable segmentations. Thus AP is not compatible with video OCL models, which produce discrete segmentations via argmax, i.e., not thresholdable.
> > >
> > >
> > >
> > > # Response to Q3: Sensitivity to the initial frame's cold-start decomposition
> > >
> > > A cascading failure is theoretically possible in recurrent systems, but empirically, Slot Attention acts as a robust attractor. Because the aggregation transform at each timestep is anchored by the current frame's visual features, the system continuously corrects itself rather than blindly reinforcing previous errors.
> > >
> > > To explicitly verify that a poor initial decomposition does not trigger a cascading failure, we conducted a targeted evaluation experiment. We took our trained models and injected heavy random noise into the slots of the very first frame during evaluation (similar to the stress test in the initial response).
> > >
> > > | @YTVIS (seed=42/43/44) |    ARI   |   ARIfg  |    mBO   |   mIoU   |
> > > |:----------------------:|:--------:|:--------:|:--------:|:--------:|
> > > |        SmoothSA        | 42.4±0.8 | 63.0±3.4 | 38.9±0.7 | 38.3±0.6 |
> > > |   + eval random corruption  | 40.5±2.2 | 60.4±2.7 | 36.6±1.3 | 36.7±0.9 |
> > > |      SlotContrast      | 37.2±0.6 | 49.4±1.1 | 33.0±0.2 | 32.8±0.1 |
> > > |   + eval random corruption  | 33.9±2.1 | 47.0±3.8 | 31.1±1.5 | 30.6±0.7 |
> > >
> > > As the results demonstrate, corrupting the initial state causes only a minor degradation in overall video metrics for both models. The sequence rapidly recovers because the recurrent SA mechanism pulls the representations back toward the visual evidence. SmoothSA maintains its lead over the baseline even when forced to recover from a corrupted initialization, confirming its robust temporal dynamics.
> > >
> > > ---
> > > [OVIS] https://songbai.site/ovis

---

### Official Review · Reviewer_su34 · 2026-03-11

**Soundness:** 3
**Presentation:** 2
**Significance:** 2
**Originality:** 3
**Overall Recommendation:** 4
**Confidence:** 3

**Summary:**

The authors propose SmoothSA to address two issues in slot attention for object-centric learning. To smooth SA iterations on the image or video’s first frame, the authors preheat cold-start queries by a tiny module self-distilled inside OCL. To smooth SA recurrences across the video’s first and non-first frames, the authors differentiate the homogeneous aggregation transforms by using full and single iterations.

**Compliance With Llm Reviewing Policy:**

Affirmed.

**Final Justification:**

My main concerns were addressed.

**Key Questions For Authors:**

1) How sensitive is SmoothSA to poor initial slot representations in preheating?
2) Could the number of SA iterations for first and non-first frames be learned adaptively rather than fixed?
3) Did you evaluate performance on non-SA-based OCL architectures to show generality?
4) Could the preheater module be replaced with a lighter alternative than a Transformer decoder without performance loss?

**Limitations:**

yes

**Strengths And Weaknesses:**

Strengths
1) Good motivation to address the issues in SA-based OCL.
2) Comprehensive experimental evaluation across synthetic and real-world image/video datasets, and downstream tasks.
3) Efficient implementation with very little overhead.

Weaknesses
1) Preheater relies on self-distilled slot representations. If initial slots are poor, preheating could propagate errors.
2) The explanation of the preheater’s attention switch is brief and could benefit from additional intuition.
3) The generalizability of the proposed method to other attention-based architectures is not explored.
4)  Lack of implementation details, which hinders the reproducibility

---

> ### Author Rebuttal · Authors · 2026-03-26
>
> # `W1` `Q1` Self-distillation Sensitivity to Poor Slot Representations
> Our designs ensure the robustness:
> - **Architectural Isolation** (`Equation 17`): We stop gradient flow between preheated queries and slots. This "firewall" disentangles our preheater from the SA module.
> - **Feature-Anchored Correction** (`Lines 184-219`): Over SA contractive iterations, any query errors are exponentially attenuated (`Equation 6`) and the final representation is anchored by features (Banach fixed-point theorem). Even in the "worst case", our preheated queries provide a sample-specific prior, which is statistically superior to image-agnostic cold-start queries.
> - **Stress Test**: We heavily corrupt the slots to test the stability: $S = S \cdot (1-\epsilon) + \mathcal{N}(0,1) \cdot \epsilon$, where $\epsilon=0.999$. We performed this 1k times evenly across 100k training steps. The negligible performance deltas verifies the robustness.
>
> | @COCO (seed=42/43/44) | MSE | ARI | ARIfg | mBO | mIoU |
> |:---:|:---:|:---:|:---:|:---:|:---:|
> | SmoothSA | 0.1350±0.0005 | 29.3±1.0  | 41.3±1.2  | 33.4±0.2  | 31.8±0.2 |
> | + random corruption | 0.1356±0.0009 | 29.6±2.5  | 41.2±1.8  | 33.0±0.7  | 31.2±0.4 |
>
>
> # `W2` Additional Intuition on Preheater's Attention Switch
> We thank the reviewer for pointing out the need for deeper intuition. We switch the self-attention (SA) and cross-attention (CA) in a Transformer decoder block as our preheater because:
> - **Information Vacuum**: Initial queries are sampled from a global Gaussian distribution, i.e., image-agnostic noise.
> - **In standard SA → CA**: Performing SA first forces non-informative queries to exchange information with each other -- The (non-linear) combination of noises is still noises.
> - **In our switched order (CA → SA)**: Performing CA first anchors non-informative queries to the input features, i.e., preheating each query with image-specific distinct object cues. Only then the subsequent SA is meaningful.
> - **Empirical evidence**: Switched order works significantly better than not (`Table 5` `Lines 357-359`).
>
>
> # `W3` `Q3` Generalization of SmoothSA to other non-SA architectures
> We focused on Slot Attention (SA) as it ***is the de facto engine for modern OCL***. Early non-SA architectures struggle on the complex real-world benchmarks (COCO, YTVIS) used in our study.
> However, we believe our "Smooth" philosophy is portable to universal bottlenecks of any iterative/recurrent refinement process:
> - **Preheating** uses a modular, plug-and-play component to solve query cold-start by providing sample-specific cues.
> - **Differentiation** balances gradient flow in recurrences across video frames.
>
>
> # `W4` Implementation Details and Reproducibility
> We appreciate the request for further implementation details. SmoothSA is designed to be straightforward and easy to integrate into mainstream OCL. Our pseudo-code is below.
> ```python
> ### for OCL on ***images***
> def forward(self, image):
>     feature = self.encoder(image)
>     query = self.initializer(num_slots)
>     query = self.preheater(query, feature)  # *** preheating ***
>         # preheater: a transformer decoder block with self-attention and cross-attention switched
>     slots, attent = self.aggregator(query.detach(), feature, num_iter=3)  # *** stop-gradient ***
>     recon = self.decoder(slots)
>     loss_recon = F.mse_loss(recon, feature)
>     loss_preheat = F.mse_loss(query, slots.detach())
>
> ### for OCL on ***videos***
> def forward(self, video, condition=None):
>     feature = self.encoder(video)
>     slots = []
>     for ti, feature_t in enumerate(feature):
>         if ti == 0:
>             query0 = self.initializer(condition if condition else num_slots)
>             query0 = self.preheater(query, feature)  # *** preheating ***
>             query_t = query0
>         else:
>             query_t = self.transition(slots_t)
>         slots_t, attent_t = self.aggregator(query.detach(),  # *** stop-gradient ***
>             feature, num_iter=3 if ti==0 else 1)  # *** differentiating ***
>         slots.append(slots_t)
>     slots = torch.stack(slots)
>     recon = self.decoder(slots)
>     loss_recon = F.mse_loss(recon, feature)
>     loss_preheat = F.mse_loss(query0, slots[0].detach())
> ```
>
>
> # `Q2` Adaptive Number of SA Iterations
> Please refer to Reviewer `gj8G` `W4`, due to limited #chars.
>
>
> # `Q4` Make Preheater Even Lighter
> We tried Cross Attention only, i.e., current preheater without Self-Attention and MLP. But it yielded inferior performance. The full block is essential for preheating, and it is already "optimally light":
> - **Training**: It introduces negligible overhead, maintaining memory and time profiles nearly identical to the RandSF.Q baseline (`Table 6`).
> - **Inference**: It adds no observable overhead in evaluation time or VRAM usage, as shown below.
>
> | @YTVIS | eval time / min | eval VRAM / GB |
> |:---:|:---:|:---:|
> | SmoothSA | 1.04 | 0.40 |
> | without preheater | 1.04 | 0.40 |
>
>
> # `*` We welcome any follow-up discussion ~

---

> > ### Author Rebuttal · Reviewer_su34 · 2026-04-04
> >
> > I have no more questions.

---

> > > ### Author Response · Authors · 2026-04-05
> > >
> > > We thank the reviewer for confirming that our responses have fully resolved all initial concerns. We are grateful for the constructive and beneficial discussion and kindly hope that the final score might reflect this positive resolution.

---

### Official Review · Reviewer_gj8G · 2026-03-13

**Soundness:** 3
**Presentation:** 2
**Significance:** 2
**Originality:** 2
**Overall Recommendation:** 4
**Confidence:** 3

**Summary:**

The paper identifies slot query code-start as a key limitation of slot attention when applied to images and the first frames of videos. To address this, the authors propose a smoothing strategy that initializes slot query with a lightweight module, and increase the number of iteration specifically for the first frame. Through extensive experiments, the paper shows the effectiveness of the proposed method across multiple tasks, including object discovery, recognition and visual reasoning.

**Compliance With Llm Reviewing Policy:**

Affirmed.

**Final Justification:**

Most of my concerns have been addressed, especially with respect to the paper’s novelty and the severity of the cold-start problem.

**Key Questions For Authors:**

- How different are the slot segmentation (in terms of redundant slots, incorrect boundaries) with and without preheating strategy?
- How severe is the cold-start problem in practice? (for example, is variance of ARI for different randomization high)? This could further validates the necessity of the proposed preheating strategy.

Minor issues: The term “differentiating” may be slightly misleading. The intent seems to be to express that the first frame and subsequent frames are treated differently, but the term “differentiating” may be confusing, as it can refer to taking gradients.

**Limitations:**

Could mention the limited novelty of the proposed method.

**Strengths And Weaknesses:**

# Strengths
- Extensive evaluation of the effectiveness of the method. The paper provides thorough experiments comparing multiple baselines and evaluating performance across a range of tasks and metrics, along with ablation studies that identifies the contributions of specific desgin choices. The experiments couldn't be more comprehensive.
- The paper points to a relatively simple but overlooked aspect that improves the performance of slot attention.

# Weaknesses
- The proposed method is conceptually straightforward, which weakens the overall contribution of this paper. In particular, the smoothing of SA recurrences merely adds more iterations to the first frame of the video. The mathematical formulation seems somewhat redundant relative to the simplicity of the idea. To add to the contribution/ novelty, additional analysis could be explored:
    + Initialize slot queries with preheater trained on object parts (e.g., a person -> head, uppder body, lower body), Such segmentations might potentially be learned in an unsupervised way. This enables studying whether initialize on object parts lead to object-part level initialization leads to object-part segmentation and controllable slot granularity. The preheater can than be used to control the granuality of the slot segmentations. In practical settings, the preheater can be conditioned on the text prompt in visual reasoning tasks, so that the slot granuality becomes task-specific.
    + Implement adaptive number of iteration steps for videos based on convergence signals (e.g., gradient magnitude like in Figure 4 right). This could better handle videos with sudden scene changes, such as objects suddenly appearing. The method can then find a balanced tradeoff between capturing dynamica scenes and reducing latency.

- Figure 1 and 2 appear highly similar. Figure 2 does not present additional information beyond what is already shown in Fig 1.

---

> ### Author Rebuttal · Authors · 2026-03-26
>
> # `W1` Simple Idea & Redundant Formulation
> We appreciate the reviewer's recognition of our method's simplicity and effectiveness.
>
> Regarding the "straightforward" nature and iteration design:
> - **Not just "adding" iterations**: SmoothSA does not simply add iterations. In ***unconditional video OCL***, baselines typically use 3 iterations for both first and non-first frames, termed "3+3". But SmoothSA ***reduces*** non-first frame iterations to 1, termed "3+1". This improves performance while reducing overhead.
> - **A Unified Principle**: In ***conditional video OCL***, baselines use "1+1". Our "3+1" strategy achieves an empirical yet elegant ***unification*** across both conditional and unconditional settings (`Lines 248-270`). The "first-frame cold start" and "non-first frame informed query" require differentiated treatment in both conditional and unconditional video OCL.
> - **Necessity of Formalism**: By analyzing Lipschitz Jacobian bounds, we formalize how SA iterations impact gradients of SA recurrences on the first and non-first frames (`Section 3.1`). Our 3+1 is specifically designed to balance these values (`Section 3.3`). ***Without such formalism, prior works naively chose*** 3+3 for the unconditional case while 1+1 for the conditional case.
>
> A "straightforward" solution that addresses a universal bottleneck with state-of-the-art results is a strength, aligned with the principle of Occam's Razor.
>
> # `W2` Additional Analyses
> We have validated our bootstrap's robustneess to poor slot representations (please see Reviewer `su34` `W1` `Q1`), and lighter preheat implementation (Reviewer `su34` `Q4`).
>
> # `W3` Controllable Granularity & Text Conditioning
> Thank you for this forward-looking insightful suggestion. We agree that controllable granularity, e.g., part-level segmentation or text-conditioned slots, is a crucial frontier for practical OCL.
>
> Controlling granularity happens in the initializer $\phi_\mathrm{n}$, which parallel works like [CTRL-O] have recently begun to explore using text prompts. However, our preheater $\phi_\mathrm{p}$ is designed to smooth SA iterations $\Phi_\mathrm{a}$, rather than re-defining the initializer.
>
> Since our current scope is mining self-supervised signals, we will highlight this promising direction in our Future Work.
>
> # `W4` Adaptive #Iterations
> Thank you for this constructive advice. We agree that dynamically adjusting #iterations could better handle sudden scene changes.
>
> As varying #iterations across a batch breaks parallelization in training, we implemented an adaptive strategy for inference.
>
> Besides, we use ***instant reconstruction loss*** as the convergence indicator rather than gradient magnitude because gradient requires an expensive backward pass.
>
> The aggregator dynamically continues iterating until the instant reconstruction loss drops below a predefined threshold.
>
> With this augmentation, SmoothSA does achieve better performance, but at slower inference speed due to calculating the indicator at each step.
>
> | @YTVIS seeds=42/43/44 | ARI | ARIfg | mBO | mIoU | throughput (videos/sec) |
> |:---:|:---:|:---:|:---:|:---:|:---:|
> | SmoothSA | 42.4±0.8  | 63.0±3.4 | 38.9±0.7  | 38.3±0.6 | 10.47 |
> | + adaptive #iters | 42.9±1.3 | 64.9±3.2 | 39.4±0.8 | 38.3±0.7 | 6.92 |
>
> # `W5` Similarity of Figures 1 and 2
> Thank you for the observation. While sharing the framework, they serve different purposes:
> - `Figure 1` (**Problem**): Shows baseline SA behavior. First-frame attention/segmentation improves ***gradually***, while non-first frames ***saturate*** quickly. This motivates our method.
> - `Figure 2` (**Solution**): Details our method ***preheating*** and ***differing***, which are highlighted with green halos.
>
> We will enhance the visual distinction to prevent confusion.
>
> # `Q1` Difference in Slot Segmentation: with / without Preheating
> Our preheating ensures sharper, semantically consistent segmentation/boundaries by stabilizing slots earlier, avoiding the fragmentation of cold-start baselines.
>
> As illustrated in `Figure 4` (left):
> - Blue row the last column (***with preheating***): Only 5/7 slots are activated to represent the sky (blue), rock (green), snow (cyan+purple, one redundant slot) and people (red).
> - Orange row the last column (***without preheating***): All 7/7 slots are activated, redundancies in rock (cyan+blue), snow (blue+red) and people (green and lime).
>
> # `Q2` Severity of Cold-Start and Variance
> Cold-start sensitivity is significant. `Table 5` `Lines 355-357` results demonstrate that preheating is necessary:
> - Drastically reduces variance (standard deviation): ARI+ARIfg 2.5 → 0.8.
> - Significantly outperforms cold-start baselines: 65.9 → 68.3.
>
> # `Q3` Misleading Term "Differentiating"
> Thank you for your kind advice. We will instead use "differ"/"differing" for clarity.
>
> # `*` We welcome any follow-up discussion ~
>
> ---
> [CTRL-O] Language-Controllable Object-Centric Visual Representation Learning

---

> > ### Author Rebuttal · Reviewer_gj8G · 2026-04-03
> >
> > I thank the authors for the rebuttal and especially their added experiments. Most of my concerns have been addressed and I'm raising the score.

---

> > > ### Author Response · Authors · 2026-04-05
> > >
> > > We thank the reviewer for acknowledging that the added experiments and clarifications addressed the concerns, and for updating the score accordingly. We appreciate the constructive reviewing and positive assessment which help us improve the clarity and completeness of our work.

---

### Decision · Program_Chairs · 2026-04-30

**Decision:**

Accept (regular)

**Comment:**

Based on three reviews and the authors’ rebuttal, I recommend **Accept**. The paper identifies a practical yet overlooked issue in Slot Attention—cold-start query instability—and proposes two simple, effective fixes: preheating queries via a light-weight module and differentiating iteration counts between first and subsequent frames. While the novelty is incremental and the theoretical contraction argument is more illustrative than rigorous, the empirical evaluation is thorough, the improvements are consistent across multiple datasets and tasks, and the rebuttal convincingly addressed concerns about robustness, reproducibility, and potential cascading failures. Overall, the work offers a solid, engineering-focused contribution that is likely to be adopted by the object-centric learning community.